# MASKED AUTOREGRESSIVE DETOKENIZATION WITH SEMANTIC VISUAL TOKENS FOR HIGH-FIDELITY IMAGE SYNTHESIS

## ABSTRACT

We propose MaDiT, a Masked autoregressive Detokenization Transformer for visual reconstruction and generation. It formulates visual tokenization as a flow-matching problem: the model learns a mapping from a standard normal distribution to the distribution of image data, conditioned on discrete visual and text tokens as well as intermediate autoregressive context. The effectiveness of MaDiT stems from two core designs. **First**, a masked autoencoder (MAE) fuses multi-modal cues from vocabulary priors and partially unmasked patterns to produce discrete visual tokens imbued with semantic meaning. This mitigates ambiguity and information loss that plague vanilla vector-quantized (VQ) representations. **Second**, we introduce a masked autoregressive de-tokenization pipeline that reconstructs images in a low- to high- frequency fashion. By initially focusing on flat, low-frequency regions and progressively refining higher-frequency details, our model reconstruct images with significantly improved fidelity. Within this pipeline, a masked decoder generates context-rich embeddings, conditioning a dedicated velocity field for precise final reconstruction. Extensive experiments show that MaDiT outperforms mainstream VQ tokenizers and enables high-fidelity visual generation on top of existing LLMs.

## 1 INTRODUCTION

Autoregressive modeling has emerged as a powerful paradigm across natural language processing Bai et al. (2023); Dubey et al. (2024) and multimodal tasks Liu et al. (2024b); Zhu et al. (2023); Yao et al. (2024), enabling unified frameworks that integrate visual perception with linguistic interfaces. Building upon this progress, an increasing number of works Fang et al. (2024); Sun et al. (2023b); Dong et al. (2023); Ge et al. (2024); Sun et al. (2024b) have begun extending autoregressive modeling for visual generation. However, most existing approaches Sun et al. (2024b); Zhou et al. (2024); Xie et al. (2024); Xiao et al. (2024) operate under a hybrid paradigm. They represent text inputs with discrete tokens as in LLMs Touvron et al. (2023),but handle visual outputs in a continuous latent space Kingma (2013); Rezende et al. (2014) by repurposing LLMs as diffusion backbones Ho et al. (2020); Song et al. (2020). This representational discrepancy introduces additional architectural complexity and computational overhead, preventing autoregressive visual generation from achieving the streamlined efficiency enjoyed by purely language tasks.

A widely adopted compromise is a two-stage VQ+diffusion pipeline for autoregressive image generation. Recent works Geng et al. (2025); Huang et al. (2025) first use an LLM to autoregressively predict a sequence of discrete image tokens, then pass these tokens as *auxiliary conditioning* to a powerful diffusion backbone (e.g., FLUX Batifol et al. (2025)) that reconstructs the final image. While this design treats image tokens analogously to text, it delegates the heavy lifting of photorealistic rendering to the diffusion decoder. Consequently, the reconstruction quality is dominated by the diffusion prior, while the discrete visual tokens act only as a weak semantic guide. For instance, in the X-Omni pipeline Geng et al. (2025) (which employs a SigLIP-VQ tokenizer with a FLUX diffusion decoder), we observe that the learned VQ token distribution is markedly more dispersed. These discrete tokens capture only coarse, high-level structure of the image, while fine-grained visual details are relegated to the diffusion process. Consequently, reconstructions from such hybrids

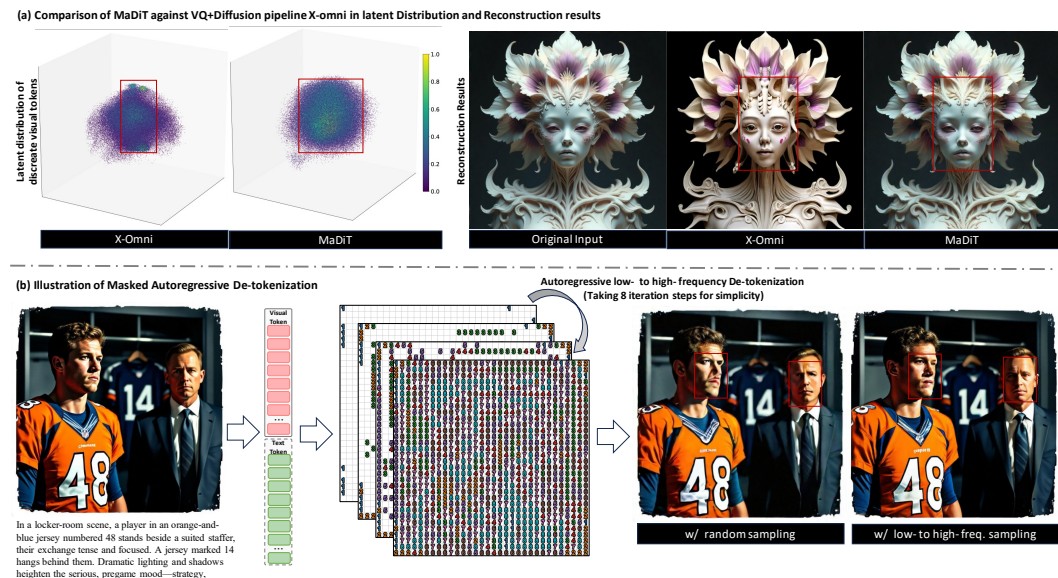

Figure 1: **Highlights of MaDiT.** (a) A hybrid VQ+diffusion baseline (X-Omni) exhibits a diffuse token distribution and produces blurred visual content that drifts toward the decoder's own learned style statistics. In contrast, MaDiT yields a more compact, uniform token distribution and preserves high-fidelity local details. (b) MaDiT's masked autoregressive de-tokenization follows a low-to-high frequency schedule guided by per-patch KL divergence. The decoder first reconstructs flat, low-detail regions; it then progressively unmasks tokens and uses earlier predictions to complete complex structures, thereby improving overall reconstruction fidelity.

often drift toward the diffusion model's own style statistics, leading to blurred high-frequency details and noticeable color shifts away from the source input, as illustrated in Figure 1 (a).

In response, we propose MaDiT – a novel masked autoregressive detokenization Transformer for high-fidelity visual generation. The effectiveness of MaDiT stems from two key designs. **First**, we incorporate vocabulary priors from a pretrained LLM (e.g., Qwen Team (2024) or LLaMA Grattafiori et al. (2024)) to infuse high-level semantic cues into the discrete visual tokens. During tokenization, MaDiT uses a masked autoencoder that fuses three sources of information: the discrete image tokens, the associated text tokens from a pretrained LLM's vocabulary, and partially unmasked patterns containing grounded visual details. By combining these cues, the encoder can alleviate the ambiguity and information loss caused by codebook collapse in standard VQ tokenization Esser et al. (2021). This results in discrete visual tokens that carry richer semantic context, providing a stronger guidance for downstream image reconstruction. **Second**, we introduce a masked autoregressive de-tokenization strategy that reconstructs the image in a low- to high- frequency manner. This approach is inspired by the information bottleneck principle of $\beta$-VAEs Higgins et al. (2017): latent embeddings with low KL divergence (close to the prior) represent flat, low-frequency regions, whereas those with high KL divergence encode complex high-frequency details. Leveraging this insight, MaDiT derives a progressive masking schedule based on per-patch KL values. The masked decoder first focuses on reconstructing the easiest content with available discrete tokens and vocabulary priors as guidance. We then iteratively unmask more tokens (reducing the masking ratio) and condition on the model's earlier outputs to gradually reveal intricate details. By reconstructing the image from low- to high-frequency content, the decoder concentrates on getting broad structures right before refining finer details, ultimately achieving higher fidelity in the final reconstruction.

We further explore two variants of the reconstruction process. (i) The decoder directly regresses the masked latent embeddings toward the target visual content at each refinement step. (ii) We build on (i) by introducing a dedicated flow-based refinement module conditioned on the decoder's output. This flow-based module models the residual details under a learned Gaussian prior and iteratively sharpens the output, further improving fidelity beyond what direct regression alone can achieve.

Bolstered by these innovations, our tokenizer achieves markedly improved reconstruction fidelity. As highlighted in Figure 1 (b), MaDiT yields a significant more compact and uniform token distribution, in stark contrast to the dispersed code distribution of X-Omni Geng et al. (2025). This indicates better preservation of fine-grained information after quantization. Consequently, our reconstructions exhibit higher local fidelity: Fine structures such as hands and facial attributes are rendered sharply and accurately. Unlike prior VQ+diffusion hybrids that often impart a stylistic bias-causing outputs to drift from the source content-our approach remains faithful to the source input.

## 2 MADIT

### 2.1 TASK FORMULATION AND OVERVIEW

**Visual Tokenization as Flow-Matching.** We formulate the visual tokenization as a flow-matching problem. The goal is to learn a mapping that transports latent variables $\epsilon \sim \mathcal{N}(0, 1)$ to samples $z$ from the visual data distribution $q$, via an ordinary differential equation (ODE). If we condition only on the discrete visual tokens, the ODE can be written as:

$$dz_t = \psi_\Theta\left(z_t, t, \mathbf{V}\right) dt \tag{1}$$

where $\psi_\Theta$ is a learnable velocity field, and $t \in [0, 1]$ is the continuous time variable, and $\mathbf{V}$ represents the discrete visual token sequence that conditions $\psi_\Theta$ to guide the generation.

Directly solving the ODE with a differentiable solver during training is computationally expensive. Instead, we regress a time-dependent vector field $u_t\left(z \mid \epsilon\right)$ whose induced dynamics trace a probability path from $\mathcal{N}(0, I)$ to the target data distribution. For optimization efficiency, we use the rectified-flow Esser et al. (2024), in which the trajectory between the target and standard normal distribution is assumed to follows a "straight-line" path:

$$u_t\left(z \mid \epsilon\right) = (1 - t) \cdot z + t \cdot \epsilon \tag{2}$$

Overall, the optimization objective for visual tokenization is to minimize the flow-matching loss:

$$\mathcal{L}_{FM} = \mathbb{E}_{t, p_t(z|\epsilon), p(\epsilon)} \left\| \psi_\Theta\left(z_t, t, \mathbf{V}^2\right) - u_t(z \mid \epsilon) \right\|_2^2 \tag{3}$$

By minimizing $\mathcal{L}_{FM}$, we learn a velocity field that can transport noise into image latents conditioned on the discrete visual tokens $\mathbf{V}$, thus achieving our visual reconstruction goals.

**Reconstruction via Masked Autoregressive Decoder.** Building on recent advances in masked generative models Li et al. (2024); Wu et al. (2025b), we employ a Transformer encoder–decoder architecture, augmented with masked image modeling, to provide strong contextual conditioning for flow-based visual reconstruction. Concretely, we partition an input image $X$ into $N$ non-overlapping patches $\{x_i\}_{i=1}^N$. We then randomly sample a masking ratio $\rho \in [0.7, , 1.0]$ (for example, $\rho = 0.7$ means 70% of patches are masked) and define the masked index set $\Omega \subseteq 1, \ldots, N$ with $|\Omega| = \lfloor \rho N \rfloor$. The Transformer encoder ingests two types of inputs to form a unified context representation: (i) embeddings of the visible (unmasked) patches, and (ii) embeddings of the quantized visual tokens $\mathbf{V}$ that represent the entire image in the learned visual codebook. To reconstruct the masked content, we introduce a learnable mask token `[m]` for each masked patch. These mask tokens are appended to the encoder's output sequence (one [m] token per $j \in \Omega$) and then passed into the Transformer decoder. The decoder produces a contextual embedding $c_j$ for each mask token `[m]`, capturing the information needed to infer patch $j$ from its surroundings.

We use these mask embeddings to condition the velocity field $\psi_\Theta$ in Equation 1. Over discretized time integration (with step size $\Delta t$), the reconstruction process for a masked patch can be factorized into a series of conditional transitions. Starting from a fully noised latent $x_t^j = \epsilon$ at time $t$, the probability of reconstructing patch $j$ back to the original $x_0^j$ under the guidance of its mask token is:

$$p\left(x_{0:t}^j \mid \text{[m]}\right) = p\left(x_t^j\right) \prod_{i=1}^t p\left(x_{(i-1) \cdot \Delta t}^j \mid z_{i \cdot \Delta t}^j, \text{[m]}, t\right) \tag{4}$$

Here, $p\left(x_{0:t}^j \mid \text{[m]}\right)$ denotes the distribution of the entire reverse-time trajectory from noised input $x_t^j = \epsilon$ to a reconstructed patch, given the conditioning mask token. Essentially, the mask embeddings $c_j$ ensure that the flow-based decoder focuses on producing a patch consistent with both the

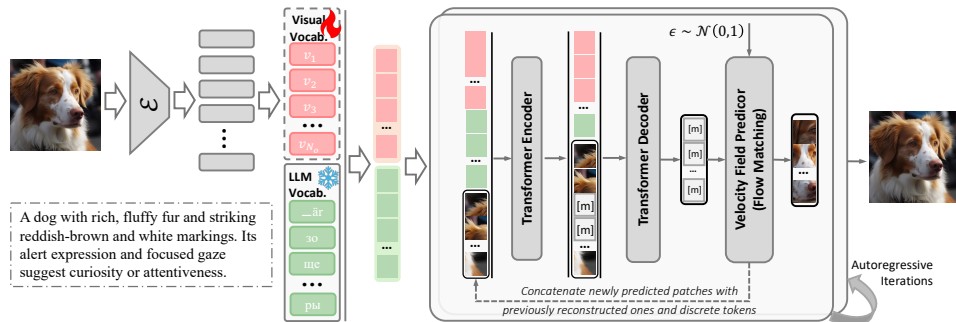

Figure 2: **Overview of MaDiT.** MaDiT first incorporates vocabulary priors from a pretrained LLM to enrich discrete visual tokens with high-level semantic cues via a masked autoencoder. This alleviates the ambiguity and information loss caused by codebook collapse. Next, a masked autoregressive de-tokenization pipeline reconstructs images through initially generating flat low-frequency structures and then progressively refining high-frequency details. A masked decoder ultimately produces context-rich embeddings that condition a dedicated velocity field for the final reconstruction.

global code sequence and the visible context, as the image is iteratively reconstructed from noise.

**Overview of MaDiT.** Figure 2 provides an overview of MaDiT, which consists of two main components: Vocabulary-enriched image tokenization (Section 2.2), which incorporates vocabulary priors to enrich visual tokens with high-level semantic cues via a masked autoencoder; Masked autoregressive detokenization (Section 2.3), which reconstruct images in a low- to high- frequency fashion.

## 2.2 VOCABULARY-ENRICHED IMAGE TOKENIZATION

We begin by encoding the input image into a continuous latent map using a pretrained VAE encoder $\mathcal{E}$ Batifol et al. (2025). This latent map is then passed through a lightweight learnable adapter (architecturally similar to Geng et al. (2025)) and flattened into a sequence of latent vectors $h_{1:L}$. Next, we apply vector quantization by replacing each latent $h_l$ with its nearest neighbor from a learned codebook $\mathcal{C} = \{c_k\}_{n=1}^{N}$. This yields a sequence of $L$ discrete visual tokens $\mathbf{V} = [v_1, v_2, \ldots, v_L]$, where each $v_l \in \mathbb{R}^d$ is the embedding for the selected codebook entry.

However, a known issue with standard VQ tokenizers is the loss of fine details Liu et al. (2025): high-frequency textures tend to get "averaged out". A limited codebook forces many distinct details to map to the same few codes, often resulting in blurry reconstructions that miss sharp textures. To mitigate this over-smoothing, we enrich the visual token representation with semantic context from a pretrained LLM's vocabulary (e.g., Qwen-2.5 Team (2024)or LLaMA-3 Grattafiori et al. (2024)). When a text description for the input image is available, we tokenize the caption and retrieve the corresponding word embeddings from a LLM. Let the sequence of text embeddings be $\mathbf{T} = [t_1, \ldots, t_S]$, where each $t_s \in \mathbb{R}^d$ after applying a linear projection to match the visual token dimension. Intuitively, $\mathbf{T}$ provides high-level semantic cues — such as object categories and attributes — that can help disambiguate visual tokens which might otherwise be confounded.

Next, we integrate these linguistic priors with the visual tokens leveraging a masked autoencoder. We concatenate the visual and text tokens to form a joint sequence $[\mathbf{V}; \mathbf{T}] \in \mathbb{R}^{(L+S)\times d}$. In addition, for each visible (unmasked) image patch during training, we include its original continuous patch embedding (the VAE encoder output prior to quantization) as an auxiliary input. Thus, the masked autoencoder processes a concatenation of all three sources of information: *(1) the quantized visual tokens* $\mathbf{V}$ *capturing global content in discrete form*, *(2) the text tokens* $\mathbf{T}$ *providing high-level semantic context*, and *(3) the original embeddings of unmasked image patches providing grounded local details*. With it, the masked autoencoder can produce a contextually enriched representation that will be leveraged by the decoder to predict masked visual patterns, as described below.

## 2.3 FLOW-BASED MASKED AUTOREGRESSIVE DETOKENIZATION

**Two-stage Training Pipeline.** We split the optimization of our masked autoregressive decoder into two sequential stages to gradually increase its reconstruction fidelity:

*Stage 1 — Latent Regression Warm-up.* The decoder learns to predict the continuous VAE latent for each masked patch, providing a coarse initialization before the flow-based refinement. We randomly select a subset $\Omega$ of patches to mask out, and treat the rest as visible context. The decoder then takes one mask token for each masked position $j \in \Omega$ and attends to the encoder's output, yielding a context embedding $c_j$. We then apply a linear projection on $c_j$ to produce $\hat{h}_j$, which is an estimate of the original latent $h_j$ for the masked patch $j$. The training objective in Stage 1 is a simple mean squared error (MSE) between the predicted $\hat{h}_j$ and $h_j$. By regressing the VAE latents of masked patches, the decoder is effectively warmed up to capture the coarse structure of the image.

*Stage 2 — High-fidelity Detail Refinement.* We refine the decoder so that it can add realistic high-frequency details to the coarse reconstructions from Stage 1. We achieve this by augmenting the decoder with a flow-based conditional MLP Li et al. (2024); Team et al. (2025) that predicts a residual update for each patch's coarse estimate. For each masked patch $j$, we take the ground-truth latent $h_j$ and add Gaussian noise to it. We then concatenate this noised latent with the coarse prediction from Stage 1. This combined vector is fed into a patch-wise MLP, which is modulated by the contextual embedding $c_j$ to output a velocity update $\Delta v_j$ for the masked patch. Intuitively, this $\Delta v_j$ denotes a direction that nudges the coarse prediction closer to the true latent, adding back the high-frequency details that were missing. We train this MLP using the flow-matching objective in Equation (3). By doing so, the model can generate textures that Stage 1 might have smoothed out.

**Low-to-High Frequency Autoregressive Sampling Schedule.** Our decoding strategy reconstructs images in order of increasing complexity – from low-frequency/simple regions to high-frequency/complex regions. This schedule is guided by the information bottleneck principle from $\beta$-VAE Higgins et al. (2017). Per-patch Kullback-Leibler (KL) divergence in a VAE indicates how much information that patch contains. Patches with large KL divergence deviate strongly from the prior, requiring more "bits" to encode and typically corresponding to texture-rich content. Conversely, patches with a small KL divergence are very close to the prior, indicating low-information regions that can be reliably predicted from surrounding context. Formally, for a single latent dimension with posterior $\mathcal{N}(\mu, \sigma^2)$, the KL divergence to the unit Gaussian prior has the closed form:

$$D_{\mathrm{KL}}\big(\mathcal{N}(\mu,\sigma^2)\,\big\|\,\mathcal{N}(0,1)\big) = \tfrac{1}{2}\big(\sigma^2 + \mu^2 - 1 - \ln \sigma^2\big). \tag{5}$$

Summing this quantity over all channels of the latent $h_i$ yields a per-patch KL score $D_{\mathrm{KL}}^i$, which measures the information content (in nats or "bits") needed to encode patch $i$. A large $D_{\mathrm{KL}}^i$ indicates that patch $i$ contains unpredictable, complex content (posterior far from the prior), while a small $D_{\mathrm{KL}}^i$ suggests the patch is low-information (posterior close to prior). Armed with this KL-based complexity measure, we guide the decoder's reconstruction order accordingly. Simpler patches (low $D_{\mathrm{KL}}$) are generated first, and more complex patches (high $D_{\mathrm{KL}}$) are filled in later. This acts as a curriculum from simple to complex content: by the time the model tackles a highly detailed area, all the surrounding simpler parts have been completed and serve as reliable context.

To implement this strategy with discrete visual tokens, we need to estimate each token's "complexity". We achieve it by maintaining an exponential moving average of the KL values for each visual codeword. For each codeword $n$, we track a running complexity score $\overline{\mathrm{kl}}_n$ that represents the average KL-term of patches quantized to the $n$-th code. This complexity score is updated incrementally during training. Formally, at each training step $t$ we update $\overline{\mathrm{kl}}_n$ as:

$$\overline{\mathrm{kl}}_n^{(t)} \;=\; (1-\alpha)\cdot\overline{\mathrm{kl}}_n^{(t-1)} \;+\; \alpha\cdot\mathbb{E}\Big[D_{\mathrm{KL}}^i\Big|\mathrm{q}(h_i) = n\Big], \forall h_i \in B^{(t)} \tag{6}$$

where $\mathrm{q}(\cdot) = n$ indicates that the latent is quantized to the codeword $n$, and $B^{(t)}$ denotes all patch latents in the $t$-th training batch. $\mathbb{E}\Big[D_{\mathrm{KL}}^i\Big|\mathrm{q}(h_i) = n\Big]$ is the sample mean of the KL-term for all patches assigned to codeword $n$ and $0 < \alpha < 1$ is a fixed smoothing factor. Over the course of training, $\overline{\mathrm{kl}}_n$ acts as a running estimate of how much information that token typically carries.

During sampling stage, we use a masked autoregressive decoding procedure similar to MAR Li et al. (2024). We begin with all patches masked and gradually decrease the masking ratio to 0 via a cosine schedule. At the first iteration, only the visual and text tokens are provided to the decoder — no image patches are visible yet. The decoder then selects a subset of patches with the smallest KL-terms to reconstruct. These might correspond to flat regions which the model can confidently generate using just the prior and global context. Once those patches are reconstructed,

we mark them as visible and feed them back into the encoder. In the second iteration, a slightly lower masking ratio is used. The decoder then reconstructs next batch of patches — again chosen by lowest remaining KL. This process repeats, revealing more patches each time, until all patches have been reconstructed. Let index the autoregressive rounds as $k = 1, 2, \ldots, K$, and $\mathbf{x}^k$ denote the set of patches reconstructed in the $k$-th iteration, the overall joint probability is factorized as:

$$p\left(\mathbf{x}^1, \cdots, \mathbf{x}^K\right) = \begin{cases} \prod\limits_{k}^{K} p\left(\mathbf{x}^k \mid [\mathbf{V}, \mathbf{T}]\right), & k = 1 \\ \prod\limits_{k}^{K} p\left(\mathbf{x}^k \mid [\mathbf{V}, \mathbf{T}], \mathbf{x}^1, \cdots, \mathbf{x}^{k-1}\right), & k > 1 \end{cases} \tag{7}$$

Here $\mathbf{x}^1$ denotes the easiest patches reconstructed in the first round (modeled conditioned only on the token sequence), and each $\mathbf{x}^k$ for $k > 1$ is generated conditioned on both the token sequence and all previously reconstructed patches. Subsequently, the masked patches produced by the decoder condition a lightweight velocity model to reconstruct the corresponding patches from noised samples. We also incorporate classifier-free guidance during sampling, by interpolating between $\psi_\Theta\left(z_t, t, [\mathbf{V}, \mathbf{T}]\right)$ and its unconditional counterpart $\psi_\Theta\left(z_t, t, \mathbf{D}\right)$ via a scaling factor $w$,

$$\tilde{\psi}_\Theta\left(z_t, t, [\mathbf{V}, \mathbf{T}]\right) = \omega \psi_\Theta\left(z_t, t, [\mathbf{V}, \mathbf{T}]\right) + (1 - \omega)\psi_\Theta\left(z_t, t, \mathbf{D}\right) \tag{8}$$

where $\mathbf{D}$ is a learnable dummy token under unconditional sampling. Overall, Algorithm 1 summarizes the sampling procedure of masked autoregressive rectified-flow decoder.

## 3 EXPERIMENTS

### 3.1 EXPERIMENTAL SETTINGS

**Evaluation Metrics.** We evaluate reconstruction on ImageNet-1K (validation split) and MJHQ at $512 \times 512$. All images are resized with aspect-ratio preservation, center-cropped, and rescaled to $512 \times 512$ before encoding. Reconstruction is assessed with four metrics: PSNR↑, SSIM↑, LPIPS↓, and reconstruction FID (rFID)↓. For text-to-image compositionality and semantic alignment, we evaluate on GenEval Ghosh et al. (2023) and DPG-Bench Hu et al. (2024). Unless otherwise stated, all metrics are reported at $512 \times 512$ under a shared preprocessing pipeline across methods. Additional implementation details, model sizes, and training schedules are provided in the Section A.3.

### 3.2 RESULTS ON RECONSTRUCTION QUALITY

**Quantitative and qualitative results.** Table 1 summarizes reconstruction on ImageNet-1K (val) and MJHQ at $512 \times 512$. Across both datasets, our tokenizer delivers consistently superior or on-par scores relative to strong VQ baselines. Notably, despite using a *smaller* codebook than Open-MAGVIT2 Luo et al. (2024) (65,536–262,144 codewords), our approach attains competitive reconstruction quality.

Figure 3 visualizes JourneyDB reconstruction results. The VQGAN-style CosMos tokenizer Agarwal et al. (2025) often misses fine facial attributes and micro-textures, yielding over-smoothed surfaces. In contrast, our masked autoregressive de-tokenization recovers high-frequency detail while maintaining global color fidelity. Relative to the VQ+diffusion pipeline X-Omni, our reconstruction results remain faithful to the source content and palette, with no systematic drift toward a decoder-specific "style.". This mitigates the softened textures and color shifts commonly observed in diffusion-conditioned hybrids.

**Ablation: de-tokenization strategy.** We ablate the impact of each decoding component in Table 2a. *(i) Direct masked regression (base).* Masked autoregressive decoding that directly regresses latent embeddings. *(ii) + Flow-based refinement.* Adding a lightweight flow-matching velocity predictor yields consistent gains in both distortion and perceptual quality (rows 1→2). *(iii) + Classifier-free guidance.* Applying classifier-free guidance (CFG) during refinement further *increases* PSNR to 23.66 and *reduces* rFID to 0.66 (row 3). *(iv) + KL-aware coarse-to-fine sampling.* Scheduling masked sampling by the offline VAE's per-region KL (low-KL first, high-KL last) focuses capacity on texture-rich regions in later rounds, reducing rFID to **0.57** (row 4).

**Ablation: vocabulary-enriched tokenization.** Table 2b considers three training settings for text–visual fusion during training: (a) *w/o vocabulary enrichment* (only discrete visual tokens condition the encoder–decoder), (b) *text to encoder only*, and (c) *text to both encoder and decoder*.

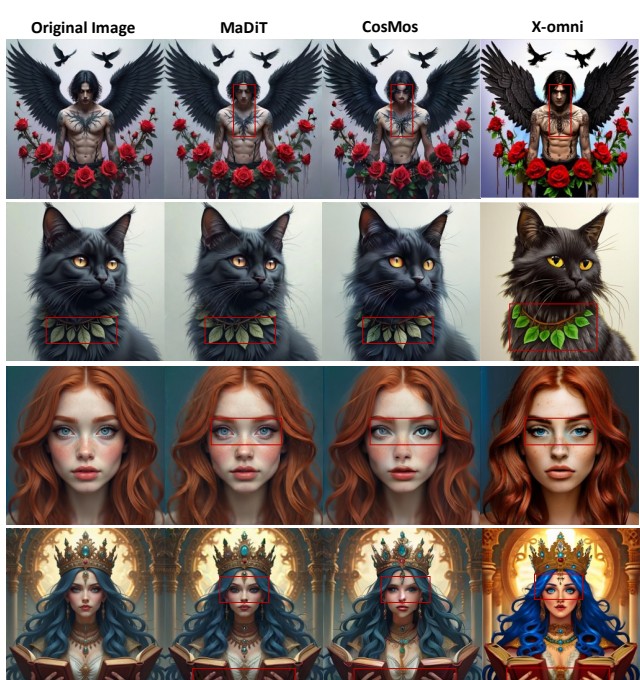

Figure 3: **Qualitative comparison of reconstruction fidelity at resolution** $512 \times 512$. We compare MaDiT with CosMos and X-Omni. CosMos (VQGAN-style) fails to recover fine facial attributes and micro-textures, producing over-smoothed surfaces. X-Omni exhibits drift toward the FLUX decoder's style, with softened textures and perceptible palette shifts. In contrast, our MaDiT can better preserve global structure while rendering crisp local detail and stable colors, remaining faithful to the source content. We also observe improved edge continuity and fewer ringing artifacts. Improvements are most evident in high-frequency regions and text details, consistent with the lower LPIPS and rFID in Table 1.

Table 1: **Quantitative reconstruction comparison** with prior VQ tokenizers on ImageNet val and MJHQ. For ImageNet val, which lacks paired captions, reconstruction is conditioned only on discrete visual tokens. For MJHQ, we incorporate multimodal conditioning using tokens extracted from text prompts and images. Ablation studies in Tables 2a and 2b follow the same settings.

| VQ-Tokenizer | Tokens | Codebook | PSNR ↑ | SSIM ↑ | LPIPS ↓ | rFID ↓ |
|---|---|---|---|---|---|---|
| **ImageNet 512×512** | | | | | | |
| Taming-VQGAN Esser et al. (2021) | $32 \times 32$ | 16,384 | 21.91 | 0.57 | 0.28 | 1.28 |
| IBQ Shi et al. (2024) | $32 \times 32$ | 16,384 | 23.36 | 0.64 | 0.23 | 0.53 |
| LlamaGen Sun et al. (2024a) | $32 \times 32$ | 16,384 | 22.51 | 0.62 | 0.24 | 0.70 |
| Open-MAGVIT2 Luo et al. (2024) | $32 \times 32$ | 262,144 | 23.84 | 0.65 | 0.22 | 0.53 |
| Cosmos-DI Agarwal et al. (2025) | $32 \times 32$ | 64,000 | 22.16 | 0.59 | 0.27 | 1.46 |
| MaDiT | 1024 | 65,536 | **23.95** | **0.65** | **0.21** | 0.57 |
| **MJHQ 512×512** | | | | | | |
| Cosmos-DI Agarwal et al. (2025) | $32 \times 32$ | 64,000 | 22.64 | 0.66 | 0.26 | 2.34 |
| IBQ Shi et al. (2024) | $32 \times 32$ | 26,2144 | 24.74 | 0.74 | 0.20 | 1.10 |
| X-Omni Geng et al. (2025) | $32 \times 32$ | 16,384 | 14.39 | 0.34 | 0.53 | 11.17 |
| MaDiT | 1024 | 65,536 | **24.90** (+10.51) | **0.75** (+0.41) | **0.20** (+0.33) | 1.04 (+10.13) |

Injecting text embeddings into the encoder-only yields the largest improvement over (a), reducing rFID by **0.26**. This indicates that semantic priors help disambiguate visually similar codes and stabilize high-frequency reconstruction.

## 3.3 RESULTS ON TEXT-TO-IMAGE GENERATION

**Quantitative results.** Table 3 reports zero-shot results on GenEval Ghosh et al. (2023) and DPG-Bench Hu et al. (2024). For GenEval, MaDiT+Qwen2.5 attains an overall score of 0.79, exceeding the autoregressive *Show-o* by +0.26 and performing on par with *Janus-Pro*. On DPG-Bench, MaDiT+Qwen2.5 achieves an overall 84.63, edging out Janus-Pro (84.19) by +0.44, and surpassing the strong diffusion baseline SD3-Medium by +0.55. These gains are consistent across categories such as entity grounding, attributes, and relations.

**Qualitative results.** Figure 4 contrasts text-to-image generations with Janus-Pro-7B Chen et al. (2025c). The visualization illustrates that our MaDiT framework, integrated with Qwen2.5-7B, better adheres to prompt semantics and maintains coherent global composition. Relative to the com-

Table 2: **Ablation study** on core components of MaDiT framework at resolution $512 \times 512$.

(a) Ablation on de-tokenization strategy at ImageNet val set.

| Latent Regression | Flow-based Refinement | CFG | KL. Sampling | PSNR↑ | rFID↓ |
|:---:|:---:|:---:|:---:|:---:|:---:|
| ✓ | | | | 22.64 | 1.03 |
| ✓ | ✓ | | | 23.02 | 0.90 |
| ✓ | ✓ | ✓ | | 23.66 | 0.66 |
| ✓ | ✓ | ✓ | ✓ | **23.95** | 0.57 |

(b) Ablation on Training setting of Vocabulary-enriched Tokenization at MJHQ dataset.

| Tokenizer | Training setting | PSNR↑ | rFID↓ |
|:---|:---|:---:|:---:|
| | w/o | 24.61 | 1.30 |
| MaDiT | encoder-only | 24.82 | 1.12 |
| | encoder-decoder | **24.90** | **1.04** |

Table 3: **Zero-shot text-to-image Generation on GenEval and DPG-Bench.** Comparison of MaDiT+Qwen2.5 with strong open-weight models.

| Generator | type | GenEval | | | | | | |
|:---|:---:|:---:|:---:|:---:|:---:|:---:|:---:|:---:|
| | | S. Obj. | T. Obj. | Count. | Colors | Position | C. Attri. | Overall↑ |
| SD3-Medium Esser et al. (2024) | Diff | 0.99 | 0.94 | 0.72 | 0.89 | 0.33 | 0.60 | 0.74 |
| MaskGen-XL Kim et al. (2025) | Mask | 0.99 | 0.61 | 0.55 | 0.81 | 0.13 | 0.31 | 0.57 |
| DC-AR Wu et al. (2025b) | Mask | 1.0 | 0.75 | 0.52 | 0.90 | 0.45 | 0.51 | 0.69 |
| LlamaGen Sun et al. (2024a) | AR | 0.71 | 0.34 | 0.21 | 0.58 | 0.07 | 0.04 | 0.32 |
| Show-o Xie et al. (2024) | AR | 0.95 | 0.52 | 0.49 | 0.82 | 0.11 | 0.28 | 0.53 |
| Janus-pro Chen et al. (2025c) | AR | 0.99 | 0.89 | 0.59 | 0.90 | 0.79 | 0.90 | 0.80 |
| OmniGen2 Wu et al. (2025a) | AR | 1.0 | 0.95 | 0.64 | 0.88 | 0.55 | 0.76 | **0.80** |
| MaDiT+Qwen2.5 | AR | **1.0** | **0.95** | **0.76** | 0.84 | 0.54 | 0.62 | 0.79 |

| Generator | type | DPG-Bench | | | | | |
|:---|:---:|:---:|:---:|:---:|:---:|:---:|:---:|
| | | Global | Entity | Attribute | Relation | Other | Overall↑ |
| DALL-E 3 Betker et al. (2023) | Diff | 90.97 | 89.61 | 88.39 | 90.58 | 89.83 | 83.50 |
| SD3-Medium Esser et al. (2024) | Diff | 87.90 | 91.01 | 88.83 | 80.70 | 88.68 | 84.08 |
| Emu3-Gen Wang et al. (2024) | AR | 85.21 | 86.68 | 86.84 | 90.22 | 83.15 | 80.60 |
| OmniGen2 Wu et al. (2025a) | AR | 88.81 | 88.83 | 90.18 | 89.37 | 90.27 | 83.57 |
| Janus-pro Chen et al. (2025c) | AR | 86.90 | 88.90 | 89.40 | 89.32 | 89.48 | 84.19 |
| MaDiT+Qwen2.5 | AR | 82.98 | **91.47** | 87.26 | **91.47** | 84.40 | **84.63** (+0.44) |

petitive method Janus-pro-7B Chen et al. (2025c), our samples show clearer high-frequency details, fewer artifacts, and more stable color palettes.

## 4 RELATED WORKS

**Visual Quantization.** VQ-VAE Van Den Oord et al. (2017) stands as a pivotal work in image quantization Lee et al. (2022); Yu et al. (2021); Peng et al. (2022). VQ-GAN Yu et al. (2021) further refines this apporach by incorporating adversarial and perceptual losses to capture more precise visual elements. RQ-VAE Lee et al. (2022) and MoVQ Zheng et al. (2022) explore multiple vector quantization steps per latent embedding. MAGVIT-v2 Yu et al. (2023) and FSQ Mentzer et al. (2023) introduce lookup-free quantization strategies, leading to large visual codebooks and expressive representations. TiTok Yu et al. (2024b) adopts a masked transformer encoder-decoder to tokenize images at resolution $256 \times 256$ into an one-dimensional sequence of 32 discrete tokens. Despite advances, the latent distributions of quantized visual tokens diverge significantly from those of text. The disparities between two modalities impose considerable challenges for autoregressive modeling. Although recent efforts Zhu et al. (2024); Yu et al. (2024a) address this by utilizing a LLM's fixed vocabulary as the visual codebook, aligning visual and linguistic modalities more directly, the resulting tokens typically preserve the intrisic two-dimensional structure of images. Consequently, autogressive models must predict visual tokens in a line-by-line manner, deviating from the one-dimensional text-processing approach used by existing LLMs.

In contrast, our proposed MaDiT uses a masked autoencoder that fuses information from multi-modal discrete tokens and partially unmasked image patches containing local visual details. By combining these cues, MaDiT alleviates the ambiguity and information loss associated with code-

Figure 4: **Text-conditioned image generation:** MaDiT (ours) vs. Janus-Pro-7B. All MaDiT samples are generated natively at $512{\times}512$. Janus-Pro renders at $384{\times}384$ and is resized to $512{\times}512$ for visualization. MaDiT produces sharper visual patterns and more reliable attribute–object bindings and spatial relations, whereas the upsampled Janus-Pro images often exhibit softened textures.

book collapse in standard VQ tokenization. This results in discrete visual tokens that carry richer semantic context, providing a stronger guidance for high-fidelity image reconstruction.

**Autogressive Visual Generation.** Most existing autoregressive visual generation models Liu et al. (2024a); Chen et al. (2025c); Team; Zhou et al. (2024); Sun et al. (2024a); Dubey et al. (2024); Tian et al. (2024) primarily focus on a sequential pixel-by-pixel process. Chameleon Team simultaneously addresses image captioning and generation within a unified Transformer framework. Janus Chen et al. (2025c) decouples visual encoding into separate pathways yet employs a single transformer for multimodal understanding and generation. Lumina-mGPT Liu et al. (2024a) captures extensive multimodal capabilities by applying a next-token prediction objective over interleaved text-image sequences. Transfusion Zhou et al. (2024) integrates next-token prediction for text with diffusion-based generation for images, unifying discrete and continuous modalities in one system. LlamaGen Sun et al. (2024a), built on vanilla autoregressive models, deliberately avoids visual inductive biases, instead advancing image generation through proper scaling. VAR Tian et al. (2024) attempts to address this concern by reframing autoregressive visual generation as a coarse-to-fine "next-scale" or "next-resolution" prediction process. However, it remains susceptible to error accumulation when predicting multiple tokens in parallel.

## 5 CONCLUSION

We presented MaDiT, an masked autoregressive detokenization Transformer for high-fidelity reconstruction and text-to-image generation. The approach combines two key components: (i) a vector-quantized tokenizer augmented with vocabulary priors from a pretrained LLM, which enriches the visual codebook without expanding its size; and (ii) a coarse-to-fine masked de-tokenization strategy guided by an offline VAE's region-wise KL profile, which reconstructs low-frequency content first and progressively refines high-frequency details. An optional flow-based refinement module further enhances local textures under a learned Gaussian prior. Integrated with a 7B-parameter backbone (Qwen2.5-7B), MaDiT achieves state-of-the-art fidelity on standard reconstruction metrics. For text-to-image generation, MaDiT attains strong results on DPGBench and GenEval.

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

## A APPENDIX

### A.1 LLM USAGE STATEMENT

We used LLMs solely as a general-purpose writing assistant. Its role was limited to language polishing (grammar, clarity, tone), line-editing and rephrasing for readability, improving section titles and transitions. All technical ideas, model designs, experiments, analyses, results, and citations were conceived, implemented, and validated by the authors. All LLMs outputs were reviewed, edited, and verified by the authors; The authors take full responsibility for all content in the paper.

---

**Algorithm 1** Sampling procedure of the masked autoregressive decoder.

---

**Require:** $[\mathbf{T}; \mathbf{V}]$ – sequence of text and visual tokens; $steps$ – total autoregressive iterations; and $\{\overline{\text{kl}}\}_{n=0}^{N}$ – an EMA average of the observed KL-term for the $n$-th codeword.

**Ensure:** $\{\mathbf{x}^k\}_{k=0}^{K}$ – reconstructed image patches (for all iterations $k$).

1:   $set \leftarrow \varnothing$                  ▷ Initialize an empty set of reconstructed patches

2:   **for** $k = 0$ **to** $steps - 1$ **do**

3:       Concatenate $[\mathbf{T}; \mathbf{V}]$ with $\{\mathbf{x}^0, \ldots, \mathbf{x}^{k-1}\}$ and feed into the masked encoder-decoder.

4:       Sample a masking ratio for iteration $k$ from a cosine schedule (starting at 1.0 and decreasing toward 0).

5:       Identify the subset of currently masked patches to reconstruct (patches with the lowest KL-term).

6:       Reconstruct the selected patches using classifier-free guidance (see Equation (8) in the main text).

7:       $set.\text{update}(\mathbf{x}^k)$           ▷ Add the newly reconstructed patches to the set

8:   **end for**

9:   **return** $set$                    ▷ Return all reconstructed patches

---

### A.2   PROCEDURE OF THE MASKED AUTOREGRESSIVE DE-TOKENIZATION

Algorithm 1 summarizes the iterative masked autoregressive de-tokenization (patch reconstruction) procedure. At the first iteration ($k = 0$), only the sequence of text and visual tokens $[\mathbf{T}; \mathbf{V}]$ is provided to the masked encoder-decoder architecture—no actual image patches are visible yet. The decoder selects a subset of patches with the smallest $D_{\text{KL}}^i$ values (i.e., the *easiest* patches) to reconstruct first. These typically correspond to flat or simple regions (e.g. sky or walls) that the model can confidently generate using just the prior and global context. Once those patches are filled in, we mark them as visible and feed their newly generated embeddings back into the encoder–decoder model for the next iteration.

In the second iteration, a slightly lower masking ratio is used (per the cosine schedule), meaning more patches are now treated as visible context. Using this expanded context, the decoder then reconstructs the next batch of masked patches—again chosen by the lowest remaining KL values. This process repeats, gradually revealing additional patches at each iteration, until all patches have been reconstructed (i.e. until the masking ratio reaches 0 and no masked tokens remain).

At each iteration, we employ the learned velocity field module to generate the pixel content for the newly selected patches. Concretely, for each selected masked patch $j$, we draw a random noise latent $\epsilon_j$ and perform rectified flow-based decoding from $t = 1$ (noise) down to $t = 0$ (signal). This sampling is conditioned on the token sequence $[\mathbf{T}; \mathbf{V}]$ as well as the patch's context embedding $c_j$ (extracted from the masked decoder). The result is a reconstructed latent for patch $j$. In this manner, the original mask token $[m]_j$ is "replaced" by a generated patch in the image. These newly generated patches are subsequently fed back into the model as known context for the following iterations. Finally, a pretrained continuous VAE then converts all predicted patch embeddings to the pixel space.

### A.3   TRAINING DATASETS AND STRATEGY

For evaluating image reconstruction on the ImageNet validation set (50k images), we train our tokenizer using only the ImageNet-1k training split, to ensure a fair comparison with prior work. For evaluating reconstructions on MJHQ-30k benchmark (a set of 30k high-quality images and captions), we assemble a diverse tokenizer training corpus from multiple open-source datasets, including PD12M Meyer et al. (2024), LAION-Aesthetic Offerman et al., LAION-Pop Lee et al. (2024), and JourneyDB Sun et al. (2023a). The resulting trained tokenizer is later integrated with the LLM for high-fidelity image generation.

For text-to-image generation, we employ a two-stage training strategy on paired image–text data. Stage 1: Pre-training for image–text alignment. We compile approximately 10 million image–text pairs from the aforementioned open datasets (PD12M Meyer et al. (2024), LAION-Aesthetic Offerman et al., LAION-Pop Lee et al. (2024), JourneyDB Sun et al. (2023a)). To enrich the textual descriptions and strengthen image–text alignment, we recaption each image using the Qwen-2.5-VL-7B-Instruct Bai et al. (2025) vision-language model, obtaining detailed descriptions averaging

80–120 tokens. Stage 2: Instruction tuning for image generation. We then fine-tune the model on high-quality instruction-following data for image generation. This includes the BLIP3o-60k dataset Chen et al. (2025a) (a curated set of 60k prompt–image pairs distilled via GPT-4$_o$) and ShareGPT-4o-Image Chen et al. (2025b), a collection of 45k text-to-image examples at GPT-4 level quality. In combination, this two-stage curriculum first equips the model with broad visual–linguistic alignment capabilities, and then refines it to produce faithful, high-quality images in response to user instructions.

### A.4 Implementation Details

Our tokenizer converts a $512 \times 512$ image into 1,024 discrete tokens, sampled from a large codebook of size 65,536. The backbone is a MAE-style masked encoder–decoder with explicit positional embeddings on all patch tokens. In our $512 \times 512$ configuration, both the encoder and decoder comprise 20 Transformer layers with hidden size 1536. During reconstruction, the decoder predicts patch-level latents, which are then transformed to RGB pixel space by the SDXL Podell et al. (2023) continuous VAE decoder.

The learned velocity-field component (used for flow-matching decoding) is modeled as a lightweight MLP with multiple residual blocks (inspired by the design in MAR Li et al. (2024)). For $512 \times 512$ images, this MLP comprises 12 residual blocks, each with 1536 channels. During the autoregressive de-tokenization stage, we perform 64 iterations (reconstructing a subset of patches in each iteration) and use an inner flow-matching sampling step size of 10 to progressively refine each patch's generation. These design choices (large codebook, deep transformer, and sufficient sampling steps) ensure that our tokenizer can capture fine-grained image details while remaining tractable in both training and inference.

To achieve autoregressive text-conditioned image generation, we employ the Qwen2.5-7B language decoder Team (2024) as our visual generative backbone. At the pre-training stage, we empirically select a global batch size of 1,024 and a learning rate of $1 \times 10^{-4}$, training the model for 10 epochs. Consistent with LLaVA Liu et al. (2023), we use the Adam optimizer without weight decay, applying a cosine learning-rate schedule with a warm-up ratio of $3\%$. For the instruct tuning stage, we adopt a learning rate of $2 \times 10^{-5}$, finetuning the model for 5 epochs.

### A.5 MaDiT Integrated with an LLM for Autoregressive Generation

Figure 5 provides an overview of how we integrate our MaDiT with a pre-trained large language model (LLM) to enable autoregressive text-to-image generation. We extend the LLM's vocabulary with a dedicated set of visual tokens corresponding to entries in the tokenizer's codebook. Specifically, we introduce special tokens <v>_0, <v>_1, <v>_2, ..., <v>_N, where $N$ is the codebook size (e.g., $N = 65,536$). We initialize these new token embeddings by drawing from a multivariate normal distribution matching the mean and covariance of the LLM's existing word embeddings. This initialization places the new visual tokens in a compatible feature space, facilitating smoother integration.

For training this coupled model, we construct single-turn instruction–response pairs using text–image data. The text prompt serves as the "instruction," and the target "response" is the sequence of discrete visual tokens (produced by our tokenizer) that represents the corresponding image. We then fine-tune the LLM on this paired data using a standard autoregressive language modeling objective (following the approach of LLaMA and related works Touvron et al. (2023); Liu et al. (2023)). Through this training, the LLM learns to output the correct sequence of visual tokens in response to a given text prompt, and to terminate the sequence with an end-of-image token once the image token sequence is complete.

At inference time, given a user's text prompt, the LLM generates a sequence of visual tokens autoregressively, until the end-of-sequence token is produced. This predicted token sequence is then passed to our MaDiT reconstruction module. Using the flow-based decoder sampling process described above, the model ultimately reconstructs a high-fidelity image corresponding to the prompt.

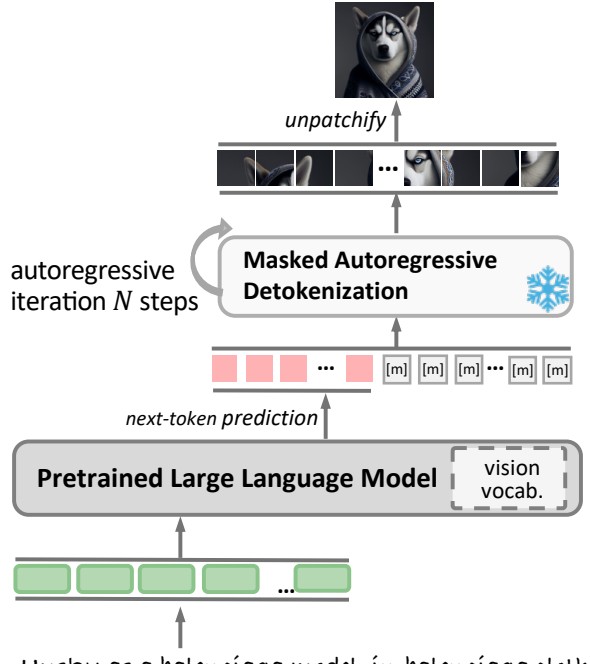

Figure 5: **Pipeline** of integrating MaDiT with LLMs for autoregressive text-conditioned generation.

## A.6    ADDITIONAL EXPERIMENTAL RESULTS

**Additional Reconstruction Results.** In Figure 6, we present supplementary 512×512 image reconstruction examples from the JourneyDB dataset. We compare our MaDiT tokenizer's reconstructions against those from X-Omni, a recent VQ–diffusion hybrid pipeline. Each pair of images (ours on the left, X-Omni on the right) highlights that our method more faithfully preserves fine details and textures from the original image. In particular, the MaDiT reconstructions maintain higher fidelity in high-frequency regions (e.g. intricate patterns, small text, and sharp edges), underscoring the effectiveness of our tokenizer's design.

**Additional Text-to-Image Generation Examples.** Figure 7 provides an extended qualitative comparison of text-conditioned image generation between our approach and the Janus-Pro-7B multi-modal model. Across a variety of example prompts, our MaDiT + Qwen2.5 method demonstrates a stronger ability to capture nuanced semantic details from the text. The images produced by our approach are often more coherent and closely aligned with the given prompt (in terms of depicted objects, attributes, and overall scene) compared to the outputs from Janus-Pro-7B. These results illustrate the advantages of our high-fidelity tokenizer and two-stage training pipeline in bridging textual semantics with visual content for generation.

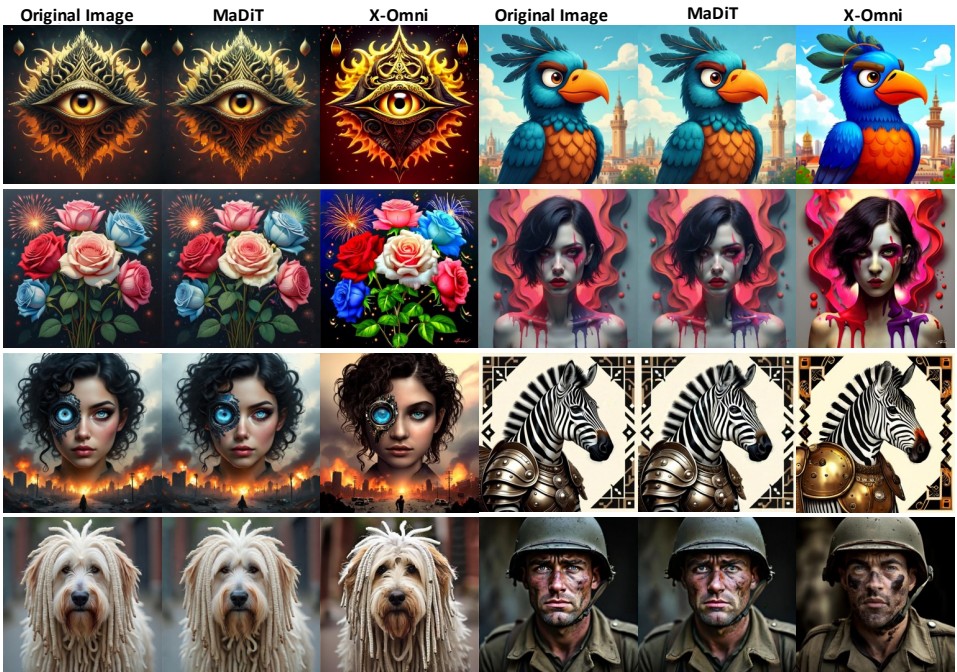

Figure 6: **Qualitative comparison** of 512×512 image reconstructions using our MaDiT tokenizer (left in each pair) versus the VQ–diffusion hybrid pipeline X-Omni (right in each pair). Our method consistently retains more fine-grained details of original images compared to X-Omni.

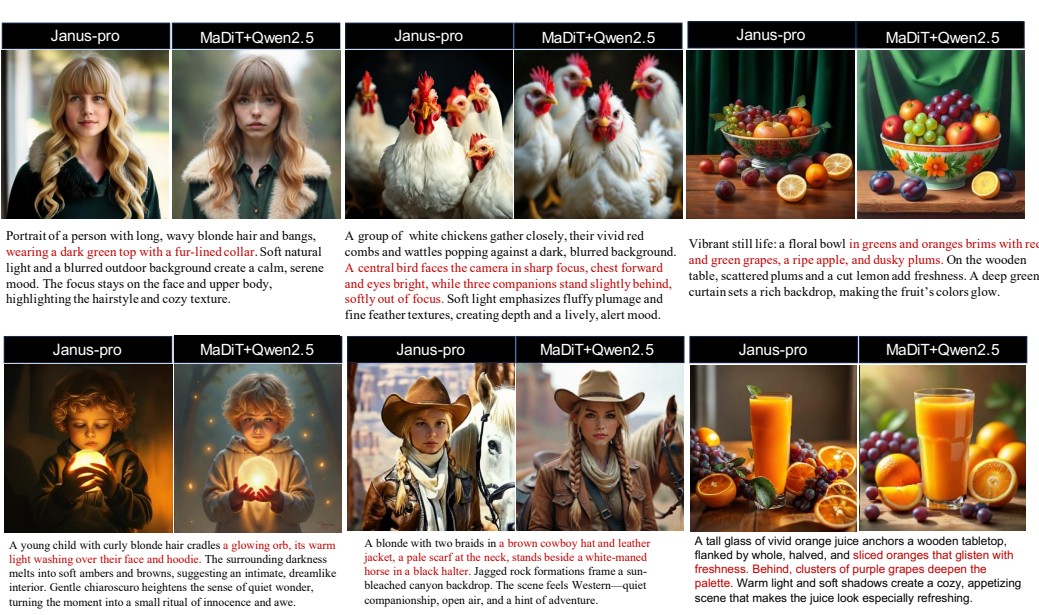

Figure 7: **Qualitative text-to-image generation results.** Comparison of images produced by the Janus-Pro-7B model (left in each pair) versus our MaDiT+Qwen25 model (right in each pair), given the same text prompts. Our generated images show greater semantic alignment with the prompts and richer details (e.g., more accurate objects and scene attributes).