# OpenReview forum: "Masked AutoRegressive Detokenization with Semantic Visual Tokens for High-fidelity Image Synthesis"
_ICLR.cc/2026/Conference — ICLR 2026 Conference Withdrawn Submission_

### Official Review · Reviewer_N3Sz · 2025-10-24

**Soundness:** 3
**Presentation:** 3
**Contribution:** 3
**Rating:** 4
**Confidence:** 3

**Summary:**

This submission proposes an encoder-decoder architecture for image generation. Specifically, the encoder encodes images & texts into discrete tokens as conditions for the decoder. Then the decoder performs image generation via masked image modeling, with the discrete tokens as semantic conditions. The decoder's generation process is trained with the flow-matching loss. Experiments show that the proposed method outperform other VQ-tokenizers in terms of PSNR, SSIM, etc.

**Strengths:**

1. The motivation is generally clear and well-explained.

2. The experiment results show the effectiveness;

3. The ablation study is generally solid.

**Weaknesses:**

1. From the method perspective, I have some concerns as follows:
    - The pipeline seems complex, and a long chain is established: encoders, quantization, masked image modeling, and flow matching. To my understanding, the core idea is adding contexts as generation conditions in the masked image modeling process. Here arises a question: why is the discrete tokenization essential? Will continuous contexts work?
    - It is evident that adding the contexts will significantly increase the computational overhead of the decoder. When comparing with other baselines, it is recommended to also report the model size and the inference cost. This is important, because the comparison would be fair when the computation is generally at a similar level.

2. The experiments.
    - How does the method compare with diffusion models?
    - It seems that the proposed coarse-to-fine decoding strategy could be applied in any masked image modeling framework.  I'm curious about its effect on the compared baselines.
    - Why isn't VAR compared? It is claimed by the authors "remains susceptible to error accumulation".

**Questions:**

I'm also curious about the potential of this method in other tasks, such as multi-modal understanding. Otherwise, I'm not sure about its advantages over diffusion models in the visual generation field.

---

### Official Review · Reviewer_f2vv · 2025-10-31

**Soundness:** 3
**Presentation:** 3
**Contribution:** 3
**Rating:** 8
**Confidence:** 1

**Summary:**

This paper proposes MaDiT for high-fidelity image reconstruction and text-to-image generation. The authors highlight two main ideas that help them derive their method: 1. vocabulary-enriched visual tokens; 2. KL-guided masked autoregressive de-tokenization. On ImageNet-val and MJHQ (512 x 512), MaDiT seems to improve PSNR/LPIPS and competitive rFID against strong VQ tokenizers with larger gains on MJHQ.

**Strengths:**

$\bullet$ Combining LLM vocabs before visual vodes and partial patch visibility is an efficient way to combat token ambiguity without exploding the codebook.

$\bullet$ KL-ordered path reconstruction is intuitive.

$\bullet$ Good performance on the empirical side.

**Weaknesses:**

Unfortunately, I cannot provide valuable insights on the weakness of this paper as my primary research area is ML theory. I have alerted PCs, ACs, and other reviewers to ignore my review. I am open to discuss theoretical questions from the authors if needed.

**Questions:**

No questions

---

### Official Review · Reviewer_y1Ah · 2025-11-01

**Soundness:** 3
**Presentation:** 3
**Contribution:** 3
**Rating:** 6
**Confidence:** 3

**Summary:**

This paper introduces MaDiT, a masked autoregressive detokenization transformer designed for high-fidelity visual reconstruction and text-to-image generation. The authors aim to address the representational gap between discrete text tokens and continuous image latents used in most hybrid AR–diffusion models. Experiments demonstrate improved reconstruction fidelity on ImageNet and MJHQ, as well as competitive zero-shot text-to-image generation on GenEval and DPG-Bench.

**Strengths:**

1. The KL-guided low-to-high frequency reconstruction offers a principled and interpretable way to schedule masking and reconstruction, improving stability and detail preservation. The optional flow module improves texture fidelity and sharpness beyond regression-only reconstruction.
2. MaDiT achieves superior reconstruction metrics (e.g., PSNR 23.95, rFID 0.57 on ImageNet) and competitive text-to-image generation, surpassing diffusion and AR baselines such as Janus-Pro and SD3-Medium.

**Weaknesses:**

1. It would be better to evaluate the proposed tokenizer on image editing tasks to further assess its reconstruction capability.
2. The integration mechanism between visual and language embeddings remains underspecified, it would help to show how LLM priors concretely affect token semantics or codebook diversity.
3. Several related literatures are missing, e.g., HART, SimpleAR. Quantitative head-to-head comparisons would help contextualize MaDiT’s gains.

**Questions:**

Please refer to the weakness part. In addition, it would be helpful to benchmark the inference speed of the proposed tokenizer.

---

### Official Review · Reviewer_9oiX · 2025-11-02

**Soundness:** 3
**Presentation:** 2
**Contribution:** 3
**Rating:** 4
**Confidence:** 3

**Summary:**

The paper proposes MaDiT, a novel visual reconstruction and generation framework designed to produce high-fidelity images from discrete visual tokens and textual prompts, while integrating seamlessly with large language models. Conventional VQ-based or VQ+diffusion pipelines — which often lose fine details, suffer from stylistic drift, and introduce architectural complexity by mixing discrete text tokens with continuous visual latents, MaDiT has the following advantages: 1) Enhances the visual tokenization stage with vocabulary priors from pretrained LLM to enrich tokens with high-level semantic cues and grounded local details. 2) Reconstructs images via a masked autoregressive, flow-matching decoder that generates content progressively from coarse low‑frequency regions to detailed high‑frequency textures. 3) Integrates directly into an LLM autoregressive generation loop, allowing the LLM to output visual token sequences in response to text, which MaDiT then reconstructs to images. They demonstrated superior reconstruction metrics against strong VQ baselines on ImageNet1K, MJHQ.

**Strengths:**

The paper proposes a novel KL-guided low-to-high frequency reconstruction strategy, using KL divergence of visual tokens to order patch generation from simple to complex, improving stability and fidelity. The MaDiT framework effectively addresses detail loss and over-smoothing in visual token-based reconstruction, and integrates smoothly with existing autoregressive text-to-image systems. The vocabulary-enriched tokenization meaningfully improves semantic disambiguation and detail preservation. The paper is well-written, with clear explanations and strong experimental results, making it a significant and applicable contribution to high-fidelity multimodal generation.

**Weaknesses:**

The experimental evaluation is limited in scope and baseline diversity. Key baselines, such as X-Omni, use SigLIP2-VQ models that are weaker than strong VAE-based tokenizers, which may overstate MaDiT advantage. Text-to-image results are only reported for Qwen2.5 + MaDiT, without showing the standalone LLM or other decoders, making it hard to evaluate MaDiT contribution. The work also lacks direct comparisons with state-of-the-art diffusion models in fine-grained generation tasks, leaving its performance ceiling uncertain.

**Questions:**

Experiments are limited. Baselines like X-Omni use SigLIP2-based VQ models, which are weaker than VAE-based methods, potentially favoring MaDiT. More comparisons with different tokenizers and a Qwen2.5-only setting (without MaDiT) are needed to validate contribution. In addition, strong diffusion models such as FLUX can achieve better fine-grained text generation and arbitrary resolution image generation; the paper should discuss MaDiT gap with such SOTA models

---

### Note · Authors · 2025-11-17

I have read and agree with the venue's withdrawal policy on behalf of myself and my co-authors.